# The Power of Food Advertisements: A Brief Mindfulness Instruction Does Not Prevent Psychophysiological Responses Triggered by Food Ads

**DOI:** 10.3390/brainsci15030240

**Published:** 2025-02-25

**Authors:** Constanza Baquedano, David Martinez-Pernia, Vicente Soto, Álvaro Rivera-Rei, Antonia Zepeda, Alejandra Vasquez-Rosati, Eugenio J. Guzmán-Lavín, Carla Ugarte, Antonio Cepeda-Benito, Vladimir Lopez, Jaime R. Silva

**Affiliations:** 1Center for Social and Cognitive Neuroscience (CSCN), School of Psychology, Universidad Adolfo Ibañez, Santiago 7941169, Chile; david.martinez@uai.cl (D.M.-P.); vicente.soto.d@uai.cl (V.S.); alvaro.rivera@uai.cl (Á.R.-R.); anzepeda@alumnos.uai.cl (A.Z.); 2Laboratorio de Fenomenología Corporal, Villarrica 4930000, Chile; alejandravasquezrosati@gmail.com; 3Centro de Investigación en Complejidad Social, Facultad de Gobierno, Universidad del Desarrollo, Santiago 7610658, Chile; eugenioguzman@udd.cl; 4Eating Behavior Research Center (CECA), School of Psychology, Universidad Adolfo Ibáñez, Santiago 7941169, Chile; carla.ugarte.p@uai.cl; 5Department of Psychological Science, University of Vermont, Burlington, VT 05405, USA; antonio.cepeda-benito@uvm.edu; 6Escuela de Psicología, Neuroscience Research Center NEUROUC, Pontificia Universidad Católica de Chile, Santiago 8320165, Chile; vlopezh@uc.cl; 7Instituto de Bienestar Socioemocional (IBEM), Facultad de Psicología, Universidad del Desarrollo, Santiago 7610658, Chile; jaimesilva@udd.cl

**Keywords:** subjective realism, mindfulness, dereification, advertised food, approach–avoidance tendencies, salivation, ERPs

## Abstract

**Background:** Exposure to visually appealing food items can enhance their subjective realism, leading to increased cravings, salivation, and automatic approach tendencies. Prior research suggests that brief mindfulness instructions promoting dereification—recognizing stimuli as transient mental events—can mitigate these automatic reactions. **Objectives**: This study assesses whether brief mindfulness instruction can mitigate automatic consumption tendencies induced by food advertisements, exploring the corresponding behavioral, physiological, and neurophysiological mechanisms. **Methods**: Sixty participants were randomly assigned to two groups: one receiving brief mindfulness instruction and the other a non-dereifying control instruction while exposed to advertised foods. This was followed by an approach–avoidance task (AAT), during which behavioral data, salivary volume, event-related potentials (ERPs) from electroencephalogram recordings, and self-reports were collected. **Results**: The results showed no significant differences in approach behaviors between the groups. Hunger, food craving, and salivation levels increased uniformly in response to food cues for both groups. The N1, N2, P3, and late positive potential (LPP) ERPs remained unaltered by the instructions and consistent with the established AAT literature. Advertising heightened the appeal of neutral foods, as evidenced by increased N2, P3, and LPP responses. **Conclusions:** The brief mindfulness instruction failed to shield participants from the automatic responses elicited by food advertising, contrasting with the effects seen with non-advertised food.

## 1. Introduction

In today’s media-driven society, advertising is ubiquitous, permeating various digital and physical landscapes in a personalized fashion. This constant exposure directly impacts both cognitive and behavioral responses in humans. Research has shown that advertising significantly influences consumers’ behaviors and shapes preferences across all age groups [1], taking advantage of evolutionarily determined neuronal mechanisms that drive implicit behavior [2]. A clear example of this is the impact of food advertisements on behavior and cognition. These ads employ multiple persuasive techniques, including vivid imagery, emotional appeals, brand associations, and multisensory cues, etc., all of which can amplify desires and shape consumer behavior [3,4]. Research has consistently demonstrated that appetitive visual food cues activate the brain’s reward pathways in a similar manner to the mechanisms engaged by addictive substances [5]. These findings suggest that visual food cues can provoke cravings, increase appetite, and promote overeating [3,6,7]. Such responses highlight the critical role of foraging in brain evolution, a function that depends heavily on visually identifying nutritious foods, especially familiar ones. This necessity may have driven the evolution of primates’ trichromatic color vision, enabling them to spot energy-rich fruits in forest canopies. This visual capacity is integral to human well-being, closely interacting with our attentional, pleasure, and reward systems, and is influenced by physiological hunger cycles. Consequently, the visual appeal of food significantly enhances the pleasure derived from eating, underlining the deep connection between visual processing and dietary choices [8,9]. For example, just reading or seeing attractive food items is enough to trigger activation in the taste and reward areas of the brain [10] or to increase salivation [11]. It has been suggested that neural responses to appetitive food cues, such as images activate similar neural networks as those involved in feeding [12].

During mental imagery, thoughts can be expressed in a way that creates a vividly realistic experience. Subjects become so deeply immersed in the content of their minds that they perceive it as real, losing awareness of the fact that they are merely imagining [13,14,15]. This mental state has been termed differently across studies, including cognitive fusion [16], reification [17], absorption [18], experiential fusion [19], imaginative immersion [15], and subjective realism [13,14]. In this way, internal food cues would activate feeding simulations that produce reward predictions whose content are gratifying and spontaneous mental events [20,21,22,23,24] about a perceived food, potentially motivating its consumption due to the somatovisceral responses that would be produced by these thoughts [12,20]. A finding that supports this theory, for example, is that, after performing food imagery exercises, salivation and more intentions to eat and buy food are reported [20,25]. In addition to the role of food imagery through different sensory modalities (imagining the smell or taste of it), it has also been associated with greater “food cravings” [20,26,27]. Previous works have shown that increasing subjective realism towards attractive foods through a brief immersion instruction increases automatic and unconscious tendencies of approach towards these foods [14,22,28]. Finally, it has been shown that people who have greater subjective realism as a trait also have greater food cravings as a trait [14], directly confirming the positive correlation between craving intensity and traits of subjective realism.

Mindfulness meditation-based interventions have emerged as a powerful tool to address the anxiety and cravings that often accompany overeating. Various studies have shown the effectiveness of this type of intervention; for example, it has been seen that, after a mindfulness-based intervention lasting seven weeks, participants who received the intervention presented significantly lower levels of food cravings than those in the control group [29]. Similarly, an eight week mindfulness-based cognitive therapy intervention led to a significant decrease in levels of food cravings, suggesting this technique is an effective way to reduce factors associated with problematic eating behaviors [30]. The authors suggested that mindfulness meditation facilitates a diminished reactivity to external cues by emphasizing an awareness of internal states of hunger and satiety [30]. To address which specific skills trained by mindfulness had the greatest impact on food cravings, Lacaille et al. (2014) [21] compared three mindfulness-based interventions. Each intervention lasted two weeks and emphasized different skills exercised by mindfulness; the first focused on “Awareness”, the second on “Acceptance”, and the third on “Dereification”. Dereification is being aware that thoughts or perceptions are merely transient representations, and not necessarily an accurate depiction of reality [17], opposing the notion of subjective realism. Dereification and related concepts have also been termed decentering [31], cognitive diffusion [16], and mindful attention [22]. The authors found that the effectiveness of mindfulness in reducing craving for chocolate was due to an increase in participants’ ability to dereify (to experience thoughts and feelings as constructed or impermanent rather than fixed or concrete) and dis-identify (to detach or separate oneself from a particular identity or set of beliefs) [21]. Moreover, studies employing brief instructions of mindful attention that directly promote dereification have shown a decrease in automatic approach impulses during the observation of attractive food items during an approach and avoidance task [14,22,28,32]. Furthermore, food elicited less salivation under mindfulness instructions (e.g., recognizing sensations and thoughts as transient mental events) compared to immersion (e.g., vividly imagine the pictured food as if it were directly in front of you) [14] and control instructions (e.g., to look at the images as you would normally do at home) [33]. These results show that the automatic approach bias towards attractive food and cephalic response preparation are dampened by mindfulness instruction. Presumably, the mental attitude of dereification reduces the perceived realism of food simulations, which in turn inhibits the formation of desires, impulses, and reflexive responses toward food [14,22,28,32].

Studies examining the neural correlates of dereification’s effect on automatic approach bias towards attractive foods [14,34] have shown that, during a brief mindful instruction, evoked brain responses—reflected in the N1 event-related potential (ERP), which is associated with efficient stimulus classification and early perceptual selection—were enhanced in incongruent trials (i.e., when participants needed to move away from attractive food stimuli) [34]. These findings suggest that, during the dereification condition, the elimination of approach bias towards attractive food could be caused by greater attention allocation and a more effective early selection [35] towards required incongruent responses (i.e., avoiding attractive food). This process may also allow for better subsequent executive processing during AAT conflict resolution. The P3 ERP component exhibited a greater amplitude in the mindful condition compared to the immersive condition, suggesting enhanced cognitive processing [36]. Additionally, approach responses elicit greater P3 amplitudes than avoidance responses, suggesting stronger motivational engagement when individuals move toward food stimuli [37,38]. Finally, for participants without previous meditation expertise, the maximum amplitude of the late positive potential (LPP) amplitudes was greater for approach than for avoidance responses. The LPP amplitude is sensitive to emotional content of images, with larger amplitudes in response to emotional rather than neutral stimuli and smaller LPP amplitudes reflecting effectively regulated responses [39].

Several studies have reported that food commercials activate reward areas associated with food intake and cravings, particularly in young people with higher body mass indexes [40,41]. It has also been observed that being aware that one’s engagement with attractive food stimuli is merely a transient mental event can reduce reward simulations and appetite behavior [28]. Considering these findings and the literature reviewed, it is concerning that, in everyday life and across various public spaces, mass media, and social networks, one is constantly exposed to cues that promote the acquisition and consumption of food, typically of low nutritional value [3,7]. Therefore, it is relevant to study the impact of advertising as a significant variable in the creation of obesogenic environments [3], and to seek ways to decrease food desire, as this plays a role in overweight and obesity.

Previous studies investigating the mechanisms by which dereification instruction modulates automatic food responses [14,22,34] used images placed on a white background without any context to ensure participants remained focused on the food and were not distracted. While this approach contributed to the internal validity of the manipulation by isolating the stimulus of interest and eliminating alternative interpretations of the effects attributed to the independent variable (mindful vs. control instructions) on participants’ reactivity to attractive and neutral food exposure, the images used had low ecological validity. In daily life, even food advertisements presented in photographs depict food within attractive contexts, often filled with color, rather than against a sterile background.

In the present study, we conducted an experiment following the approach outlined by Papies et al. (2012) [22] and Baquedano et al. (2017, 2019) [14,34], but with a key modification: we used advertised food items instead of the plain white backgrounds utilized in their research. Therefore, to explore whether dereification instruction could be a tool to help resist the impact of food advertisements, experiments need to take a further step and attempt to replicate the findings using images with greater ecological validity. If successful, the results of this study could provide evidence supporting the development of brief, targeted interventions for mass media and social network users exposed to food advertising.

We used an approach and avoidance paradigm with food images, where a dereification instruction was given while adding backgrounds with colors and shapes to make the images more similar to actual food advertisements. The aim was to determine whether adopting a dereification attitude could counteract automatic consumption tendencies such as salivation, approach behaviors, and cravings induced by food advertisements, while incorporating brain and physiological measurements to explore the brain mechanisms potentially influenced by dereification instructions on food cravings elicited by advertising. We hypothesize that food advertising, akin to an immersion instruction in food items, will increase food cravings and automatic approach tendencies towards attractive food images under a control instruction. Conversely, the application of a brief mindful instruction promoting dereification during food advertising will decrease and neutralize food cravings and automatic approach tendencies towards attractive food images, as well as salivation elicited by the advertising. In the mindful condition, during the approach and avoidance task, there will be a greater amplitude of evoked potentials N1 and LPP in trials involving moving away from attractive food versus approach trials. The brief mindful instruction, operating over the advertising, will require the recruitment of a larger number of neural networks for attention, inhibition, and reevaluation to correctly respond in the AAT to the advertised foods.

The paper follows a structured format, beginning with the Methods section, which outlines the experimental design and procedures. The Results section then presents key behavioral and physiological findings, followed by the Discussion, which analyzes our results in comparison with the previous literature and their implications for food advertising and mindfulness. The paper concludes with separate sections for Conclusions, Limitations, and Future Directions, summarizing key insights, addressing constraints, and proposing avenues for future research.

## 2. Materials and Methods

### 2.1. Participants and Recruitment

We recruited university students who responded to social media messages sent to student groups and organizations. Respondents with conditions that could either confound the results or for whom participation could pose a risk to their health were excluded from participation. The exclusion criteria included self-reported histories of mental health disorders or cardiac conditions, as well as having a BMI > 30 or <18.5. All participants received USD 15 to help offset transportation costs to the laboratory. The final sample included 60 college students (42 females and 18 males), aged *M =* 22.55, *SD =* ±2.21 years, with a diverse range of dietary preferences, including 10 vegetarians and 3 vegans. The participants were randomly assigned to two experimental conditions. The Mindful instruction group comprised 29 participants (23 females and 6 males), aged *M =* 23.14, *SD =* ±2.38 years, of whom 4 were vegetarians and 1 was vegan. The Control instruction group included 31 participants (19 females and 12 males), aged *M =* 22.00, *SD =* ±1.91 years, with 6 vegetarians and 2 vegans. All procedures in this study were approved by the Ethics Committee of the University of Santiago de Chile (protocol code No. 010/2023), in accordance with the guidance and regulations of the National Agency for Research and Development (ANID) of Chile. All participants gave informed consent prior to participating in the study.

### 2.2. Procedure

Participants were instructed to have their last regular meal at least two hours prior to arriving at the laboratory. Upon arrival, they completed a pre-state questionnaire assessing baseline levels of Hunger and Food cravings. Then, the electroencephalogram (EEG) was installed. Once the EEG was in place, the first saliva sample was taken (sample T0), and the experiment began with an ‘Exposure’. In this phase, participants were first instructed to adopt a specific mental state (mindful or control) through a video while being presented with 60 images, which they were to observe with the attitude requested in the video. This instruction could be a Control instruction (i.e., watch the images as if you were watching a random TV program) or a Mindful instruction, aiming to produce a dedicated attitude (i.e., consider the sensations evoked by the images as transient mental events). After this stage, the second saliva sample was taken (sample T1), and then participants were asked to complete the approach and avoidance task (AAT) with the images previously shown in the ‘Exposure’. During the AAT, participants were instructed to maintain their focus on the screen and respond as quickly as possible to visual cues. After this stage, the last saliva sample was taken (sample T2). Once the third saliva sample was collected, participants repeated the same sequence, i.e., Exposure and AAT, with the remaining 60 images. This division of the experiment into two halves was intended to reduce monotony and enhance participant engagement. EEG recordings were obtained during both the exposure and the approach–avoidance tasks. At the end of the session, participants filled out the post-state questionnaire (see below for details) and were weighed and measured to obtain their BMI (Figure 1). Finally, participants were sent a set of trait questionnaires by mail to complete at home after the experimental session, which included food preference questionnaires.

### 2.3. Stimuli

We used a subset of 120 food images (60 attractive and 60 neutral) from the Food-pics image database [42], which was validated for the Chilean population by Baquedano et al. (2017, 2019) [14,34]. The images were designed to imitate advertising in terms of design and color, but did not include letters or numbers. Stimuli were adjusted for color and luminance using Fiji, an open-source image editing tool, ensuring that the neutral and attractive image pools exhibited no significant differences in RGB values or luminescence parameters that could influence early EEG ERPs. Following the laboratory session, participants were asked to rate the attractiveness of each image at home according to their attractiveness on a 5-point Likert scale (from “5—very attractive” to “1—not attractive”). These data were used to create personalized pools of 50 attractive (highest score) and 50 neutral (lowest score) food images for each participant, which were then used in the reaction time (RT) and EEG analysis.

### 2.4. Approach–Avoidance Task (AAT)

We implemented a protocol that involves exposure to neutral and attractive food images, incorporating either mindful or control instructions, followed by an approach–avoidance task (AAT), as outlined by Papies et al. (2012) [22] and Baquedano et al. (2017, 2019) [14,34].

More specifically, during the ‘Exposure’ phase, participants were instructed to adopt a specific mental state—either mindful or control. In the mindful condition, participants were directed to “be aware of thoughts, sensations, and reactions while viewing each image, considering these mental events as mere transient mental constructions”. In the control condition, they were instructed to “look at the images as they would naturally do, for example, in front of the television”. (A detailed instructions transcript and the video links can be found in Appendix A). They were then presented with food images for 5 s each, which they were required to observe in accordance with the attitude directed in the instructional video.

Subsequently, participants were asked to complete the approach–avoidance task (AAT) with the images previously shown in the ‘Exposure’ phase. In these AAT sessions, participants were presented with food images that were either attractive or neutral, overlaid with a blue circle or square to signal whether they should pull toward or push away from the image. Responses were made using a joystick; pulling it signified an approach, while pushing indicated avoidance. Once the joystick was moved beyond a certain threshold, the images would either increase or decrease in size to mimic movement toward or away from the viewer. Participants were given a 2000 ms window to respond. If no action was taken, the next image appeared after 2000 ms or between 650 ms and 1050 ms following a response. Each of the 120 images was displayed four times during a single AAT block, resulting in a total of 240 trials per block and 480 trials across both conditions. The experiment was structured to include 120 trials for each of the following: approaching attractive images, avoiding attractive images, approaching neutral images, and avoiding neutral images, across the different setups. Researchers instructed participants to maintain their focus on the screen and respond as quickly as possible to visual cues.

### 2.5. Self-Reports Instruments and Trait Measurements

**Pre-State and Post-State Questionnaires**: The pre-state questionnaire consisted of the following two questions assessed on a 7-point Likert scale: “Are you hungry at this moment?” and “How strong is your desire to eat right now?” where 1 indicated “Not hungry at all” and 7 indicated “Extremely hungry”. This process enabled us to confirm retrospectively that both groups had comparable levels of hunger and food cravings before the experiment began.

In addition to including the two pre-state items, the post-state questionnaire consisted of 16 items that measured four critical dimensions of subjective realism. The dimensions are ‘Craving’ and ‘Stickiness’, which, respectively, assessed the impulsive and motivational impact of perceiving food images as real, and ‘Dereification’ and ‘Meta-awareness’, which indexed different but related dimensions of how real the participants’ experiences of food were during the experiment. An overall score for this post-state questionnaire was calculated by averaging the scores across these dimensions. For more details, additional examples, and the validation of this questionnaire, please refer to Appendix A [13].

**Trait Questionnaires and Measurements:** In order to control relevant trait characteristics between groups or establish relationships between trait characteristics and the Instruction condition effects in the AAT, a battery of self-report questionnaires was applied after finishing the experimental lab setting. This battery was composed of the following questionnaires: the Power of Food Scale (PFS), a psychometric tool designed to assess the psychological impact of living in food-abundant environments [43]; the external eating subscale of the Dutch Eating Behavior Questionnaire (DEBQ), which quantifies the propensity of individuals to eat in response to external food-related cues [44,45]; and the Dereification-as-a-Trait questionnaire [14]. Additionally, we weighed and measured participants to calculate their body mass index (BMI).

### 2.6. Behavioral Measures

Reaction times (RTs) were recorded during the AAT, capturing the interval from when the food image with the cue appeared on the screen to when the participant responded.

### 2.7. Physiological Measures

**Electrophysiological Signal Acquisition and Preprocessing:** Electrophysiological recordings were conducted during both the Exposure and the AAT phases under the two experimental conditions (Control and Mindful). We employed a 128-channel Biosemi ActiveTwo digital EEG system. The central electrode of the cap was positioned on the participants’ heads and aligned to anatomical landmarks on the scalp, namely the midpoint between the nasion and the inion, as well as the midpoint between both preauricular points. The sampling rate for recording was initially set at 1024 Hz, but the data were subsequently down-sampled to 512 Hz during preprocessing. BDF files were preprocessed in Matlab (Version: 9.14.0.2489007 (R2023a) Update 6) using the EEGLAB 2023.1 toolbox [46] and the ERPLAB 10.0 toolbox [47], and were automated using in-house scripts. EEG data were high-pass filtered at 0.1 Hz using an IIR Butterworth filter with a 12 dB/oct roll-off. A working time window was selected from 2 s before the first stimulus to 2 s after the last one. Defective channels were identified using eye inspection and omitted from the preprocessing steps. Artifacts originating from eye blinks or movements, heart beats, muscle contraction, channel noise, or electrical interference were identified and removed by means of independent component analysis (ICA) with the Extended Infomax algorithm (runica) for ICA decomposition, with no PCA dimension reduction and using the ICLabel 1.4 classifier [48]; additionally, components with an IC class probability score [48] between 0.8 and 1 in the aforementioned artifactual categories were removed. At this point, defective channels were spherically interpolated subsequent to ICA to retain a full rank for ICA [49]. Data were re-referenced to infinity, using the three-concentric-sphere head model with the REST 1.2 toolbox [50], low-pass filtered at 30 Hz using an IIR Butterworth filter with a 12 dB/oct roll-off, cut in [−200 ms 900 ms] epochs around stimuli, and baseline-corrected relative to the mean voltage in the pre-stimulus time. An automatic artifact rejection system excluded any segments with amplitude fluctuations exceeding ± 100 μV. During the recording period, 5 EEG recordings were deemed too noisy and artifact-ridden to be useful; therefore, they were replaced with fresh participants. Despite the artifact rejection, a sufficient number of artifact-free segments remained for all 60 participants across each of the following image–response combination: Avoid Neutral (*M =* 94.03, *SD =* ±6.69); Approach Neutral (M = 93.78, *SD =* ±7.00); Avoid Attractive (*M =* 93.66, *SD =* ±6.93); and Approach Attractive (*M =* 94.1, *SD =* ±6.27). There were no significant differences in the count of usable trials post-artifact rejection among different groups or trial types.

**Event-Related Potentials (ERPs):** Given our previous results, we examined the N1 and LPP, which have previously been shown to be sensitive to mindful instructions in AAT paradigms [34]. Additionally, we assessed the N2 and P3 ERP components, as these ERPs have been widely modulated by subjective emotional valence within AAT conflicts across several studies [51,52,53,54]. The N1, N2, P3, and LPP were split across three successive time windows and were evaluated based on visual inspection of the stimulus-locked grand average ERP waveforms. These were quantified at electrode sites chosen according to topographical and amplitude maps and corroborated in the previous literature [55]. Epochs were then averaged for each of the four combinations of picture and response type. The N1 ERP was analyzed within the 145 to 175 ms timeframe over the occipital region, including electrodes O1, O2, and Oz [56,57]. The N2 ERP was analyzed between 220 and 270 ms at the C1, Cz, and C2 electrodes. The P3 amplitude was analyzed between 320 and 450 ms at the P1, Pz, and P2 electrodes [53,54,58]. LPP was assessed within the 450 to 550 ms timeframe at the parieto-occipital ROI (including PO7, PO3, PO8, and PO4); LPP2 was assessed within the 550 to 650 ms timeframe at the parietal ROI (including P1, Pz, and P2); and LPP3 was assessed within the 650 to 750 ms timeframe at the central ROI (including C1, Cz, and C2). These ROIs were selected on the basis of amplitude topographical maps and previous literature reports (see a similar approach in Bamford et al., 2015) [51]. Amplitudes for these components were calculated as the average voltage within the above-mentioned windows.

#### Saliva Collection

We evaluated whether, in line with the literature, the presentation of food cues triggered cephalic phase responses (CPRs), (i.e., an increase in salivation) [11]. Saliva samples were collected at the following three points during the experimental sequence for each condition: prior to the start of the experiment (‘T0′, baseline sample); following the exposure phase (‘T1′ sample); and after the subsequent AAT block (‘T2′ sample) (Figure 1). Saliva collection involved passive drooling [59] into a 5 mL cryotube for one minute, immediately followed by storage at −20 °C. The volumes were measured using an Eppendorf micropipette (p1000).

### 2.8. Statistical Analysis

All statistical analyses were performed in R (version 4.4.2 (2024-10-31)) (R Core Team, 2023).

#### 2.8.1. Sample Size

The required sample size was calculated based on an expected effect size in behavioral and EEG measures from prior studies [34], with reported effect sizes ranging from f = 0.21 to f = 0.25. Using a mixed-design ANOVA with two independent groups and two repeated measures, a desired power of 0.80, and an alpha level (α) of 0.05, a minimum sample size of 48 participants was required to detect an effect size of f = 0.21. To account for an anticipated 10% dropout rate and potential technical issues during data collection, we planned to recruit a total of 60 participants. The calculation was conducted using G*Power 3.1 [60].

#### 2.8.2. Reaction Times (RT) Analysis

We conducted a repeated-measures ANOVA, using the Greenhouse–Geisser correction to account for violations of sphericity. The analysis included the Instruction group (Control/Mindful) as a between-subjects factor and the Food type (Neutral/Attractive) and Response type (Approach/Avoidance) as within-subjects factors. The ANOVA was performed using the afex package [61], which provides robust tools for factorial experimental designs. Post hoc pairwise comparisons were conducted using the emmeans package [62] with Tukey’s honest significant difference (HSD) adjustment to control for Type I error rates.

#### 2.8.3. Salivary Volume

We conducted a two-factor repeated-measures ANOVA (rmANOVA), using the Greenhouse–Geisser correction to adjust for violations of sphericity. The model included the Instruction group as a between-subjects factor and the Sampling Time (T0/T1/T2) as a within-subjects factor. The analyses were performed using the afex package [61], which provides robust methods for factorial designs. Post hoc pairwise comparisons were conducted using the emmeans package [62], with Tukey’s honest significant difference (HSD) adjustment applied to control for Type I error rates.

#### 2.8.4. Self-Report Analysis

To assess the impact of Instruction group and Time on participants’ self-reported ratings of Hunger and Food Craving, a two-way ANOVA was employed. The analysis used Instruction group as a between-subjects factor and Time (Pre/Post) as a within-subjects factor. Post hoc comparisons were carried out with the emmeans package [62], applying a Tukey’s HSD test for multiple comparisons. To analyze the outcomes of the post-state questionnaire, we employed paired *t*-tests to evaluate the differences in responses between the instruction groups for each sub-dimension—namely, dereification, meta-awareness, craving, and stickiness—as well as for the overall score. This approach allowed us to systematically assess the impact of each instructional condition on these specific dimensions of subjective realism towards perceived food images. We utilized paired *t*-tests to analyze each trait questionnaire (i.e., the Power of Food Scale (PFS), the external eating subscale of the Dutch Eating Behavior Questionnaire (DEBQ), and the Dereification-as-a-Trait questionnaire) as well as body mass index (BMI). This approach was adopted to control for relevant trait characteristics that might influence responses in the approach–avoidance task (AAT) across different instruction groups, thereby isolating the effect of the instruction itself. Bonferroni corrections were used for all *t*-tests.

#### 2.8.5. Event-Related Potential (ERP) Analysis

To assess the effects of group, food, and response on ERP amplitudes, we utilized a linear mixed-effects model (LMM), fitted with restricted maximum likelihood (REML) as implemented in R with the lme4 package, using the lmer function [63]. The actual model used was as follows:**amplitude ~ group * response * food + (response * food|num_id) + (1|num_id:electrode)**
Instruction group, Response type, and Food type (and their interactions) were included as fixed effects. Also included were random intercepts and slopes for Response and Food (and their interactions) for each participant (num_id). Finally, random intercepts were allowed to vary by electrode nested within participant. To assess the significance of each fixed effect and their interactions, an ANOVA with type III sum of squares was performed on the *lmer* model. For each significant effect, a posteriori tests were performed using the *emmeans* package (Kenward–Roger df correction). Also, the partial omega squared (ω^2^) was calculated as a measure of effect size (package *effectsize*; [64]).

## 3. Results

### 3.1. Self-Reports and BMI

**Trait Questionnaires and Measurements**: Welch’s two-sample *t*-tests were used to assess differences between the two Instruction groups across several trait questionnaires and measurements, with no significant differences found (Table 1). Given that all trait measures were equivalent between the two Instruction groups, these trait variables are unlikely to explain the group differences, other than the type of instruction provided, on the dependent variables of the experiments.

**State Measurements:** The effect of Instruction group and Time on participants’ self-reported Hunger and Food Craving was evaluated using a two-way ANOVA, using Instruction group (Control/Mindful) as between-subject factor and Time (Pre vs. Post) as within-subject factor. For Hunger, the ANOVA results indicated a significant main effect of Time on hunger scores (F(1, 58) = 73.744, *p* < 0.00001, η^2^ = 0.271). The main effect of Instruction group on hunger scores was not statistically significant (F(1, 58) = 2.367, *p* = 0.129, η^2^ = 0.028) and the interaction between Instruction group and Time was also not significant (F(1, 58) = 0.287, *p* = 0.594, η^2^ = 0.0014). These results indicate that the level of hunger was not different at the baseline and that experimental sessions increased participants’ hunger, unaffected by the Instruction group type (see Figure 2a). For Food Craving, the ANOVA results revealed a significant main effect of Time on Food Craving scores (F(1, 58) = 66.740, *p* < 0.00001, η^2^ = 0.257). The main effect of the Instruction group on Food Craving scores was not statistically significant (F(1, 58) = 2.851, *p* = 0.097, η^2^ = 0.033). Food Craving scores were significantly higher post-session compared to pre-session across both Instruction groups. The interaction effect between Time and Instruction group was also not significant (F(1, 58) = 0.127, *p* = 0.723, η^2^ = 0.00066). This indicates that food craving was not different at baseline and that the experimental session’s effect on increasing the desire to eat was consistent across both Instruction groups (see Figure 2b). Additionally, we used a post-state questionnaire to determine whether the type of instruction impacted self-report measures of subjective realism, and to confirm that subjects properly followed the instructions. Furthermore, we employed a post-state questionnaire to assess the impact of the type of instruction on self-reported measures of dimensions contributing to subjective realism. This also helped confirm that participants correctly applied the instructions to the food images. Our analysis, conducted using Welch’s two-sample *t*-tests, revealed that, contrary to our hypothesis, neither the global score nor any of the subscales were modulated by the Instruction group factor; additionally, neither was *global score* (*t*-test, t(57.319) = 0.08754, *p* = 0.9305); *dereification* (*t*-test, t(55.42) = 1.6804, *p* = 0.0985); *meta-awareness* (*t*-test, t(57.622) = −1.1612, *p* = 0.2503); *craving* (*t*-test, t(56.518) = 1.0286, *p* = 0.308); or *stickiness* (*t*-test, t(55.106) = 0.66718, *p* = 0.5074).

### 3.2. Behavioral Results

Regarding AAT reaction times (RTs), we hypothesized that we would find behaviors consistent with the findings of Papies et al. (2012) [22] and Baquedano et al. (2017) [14] when the control and mindful instructions were given to each subject. To test these hypotheses, we conducted a repeated-measures ANOVA, with Instruction group as a between-subject factor and Food type and Response type as within-subject factors. The results of this ANOVA showed a main Response effect (F (1, 58) = 14.08, *p* < 0.001, η^2^G = 0.06), where reaction times were significantly faster for approaching food items (*M* = 590, *SD =* ±12.0) than avoiding them (*M* = 604, *SD =* ± 12.2). The contrast between these cues was statistically significant (estimate = 13.8 ms, SE = 3.67, t(58) = 3.752, *p* = 0.0004), indicating a consistent effect across Instruction groups and Food type conditions. No interaction was found with Food type or Instruction group, meaning this response effect was the same for both types of food (attractive and neutral) and for both instructions (see Figure 3a).

### 3.3. Salivary Results

Saliva volume data were analyzed through a mixed rmANOVA including Instruction group (Mindful and Control) as between-subjects factors and Sampling Time (T0, T1, T2) as a within-subject variable. This rmANOVA showed an increase in salivary volumes over time for both groups (F (1, 58) = 13.66, *p* < 0.001, η^2^G = 0.05). T0 < T1, *p* = 0.03 and T1 < T2, *p* = 0.002 (Figure 3b).

### 3.4. Electrophysiological Results: ERP During the Approach–Avoidance Task

We employed a linear mixed-effects model (LMM) to assess the modulation of ERP amplitudes based on the provided instructions. Our hypothesis posited that AAT-related ERP modulations would interact with the type of instruction given (Instruction groups). Separate models were created for each ERP component. There were no significant effects involving the Instruction groups. The full reports for each ERP can be found down below.

#### 3.4.1. N1 ERP

The analysis of variance (ANOVA) applied to the LMM for the N1 component revealed a significant effect of Food type (F (1, 58) = 51.895, *p* < 0.001, ω^2^ = 0.46) and Response type (F (1, 58) = 7.131, *p* = 0.010, ω^2^ = 0.09). For Food type, the post hoc tests indicated that the amplitude for attractive food (*M* = −0.826, *SD =* ±0.559) was significantly greater than that for neutral food (*M* = −0.088, *SD =* ±0.547; t.ratio(58) = −7.204, *p* < 0.001). For Response type, the post hoc tests indicated that the amplitude for the approach response (*M* = −0.609, *SD =* ±0.558) was greater than that for the avoidance response (*M* = −0.305, *SD =* ±0.549; t.ratio(58) = 2.670, *p* = 0.010) (Figure 4a).

#### 3.4.2. N2 ERP

The analysis of variance (ANOVA) applied to the LMM for the N2 component revealed a significant effect of Response type (F (1, 58) = 8.285, *p* = 0.006, ω^2^ = 0.11). The post hoc tests indicated that the amplitude for the avoidance response (*M* = −4.33, *SD =* ±0.317) was significantly greater than that for the approach response (*M* = −4.06, *SD =* ±0.303; t.ratio(58) = −2.878, *p* = 0.006). (Figure 4b). These findings align with previous research on N2′s role in conflict monitoring and resolution, supporting the idea that N2 amplitude increases with cognitive effort required to resolve approach–avoidance conflicts.

#### 3.4.3. P3 ERP

The analysis of variance (ANOVA) applied to the LMM for the P3 component revealed a significant effect of Response type (F (1, 58) = 4.548, *p* = 0.037, ω^2^ = 0.06). The post hoc tests indicated that the amplitude for the approach response (*M* = 1.90, *SD* = ±0.339) was significantly greater than that for the avoidance response (*M* = 1.66, *SD* = ±0.352; t.ratio(58) = −2.133, *p* = 0.037) (Figure 4c).

#### 3.4.4. LPP1 ERP

The analysis of variance (ANOVA) applied to the LMM for the LPP1 component revealed a significant effect of Food type (F (1, 58) = 51.895, *p* < 0.001, ω^2^ = 0.46) and Response type (F (1, 58) = 7.131, *p* = 0.010, ω^2^ = 0.09). The ANOVA for the LPP1 component revealed a significant effect of Response type (F (1, 58) = 6.137, *p* = 0.016, ω^2^ = 0.08). The post hoc tests indicated that the amplitude for the avoidance response (*M* = 3.34, *SD* = ±0.406) was significantly greater than that for the approach response (*M* = 2.97, *SD* = ±0.462; t.ratio(58) = 2.477, *p* = 0.016) (Figure 5a).

#### 3.4.5. LPP3 ERP

The analysis of variance (ANOVA) applied to the LMM for the LPP3 component revealed a significant effect of Response type (F(1, 58) = 7.925, *p* = 0.007, ω^2^ = 0.10). The post hoc tests indicated that the amplitude for the avoidance response (*M* = −1.125, *SD =* ±0.297) was significantly greater than that for the approach response (*M* = −0.702, *SD =* ±0.287; t.ratio(58) = −2.815, *p* = 0.007) (Figure 5b).

## 4. Discussion

In this study, we aimed to replicate the effects of a brief mindfulness instruction compared to a control instruction on responses to attractive and neutral food stimuli, with a key modification: incorporating colored backgrounds into the food stimuli to make them more similar to food advertisements. Based on prior findings [14,22,28], we hypothesized that the mindful instruction would mitigate salivation, automatic approach tendencies, and food cravings in response to the advertised food stimuli. Furthermore, we sought to explore the neurocognitive mechanisms underlying this potential modulation in participants without prior meditation experience. Specifically, we predicted that neurological indicators would differentiate between neutral and attractive food images and that the brief mindfulness instruction would modulate neurophysiological responses indicative of disruptions in automatic approach–avoidance responses to attractive stimuli.


**Behavioral and Physiological Measures**


Contrary to our expectations, reaction time data from the primary task, the approach–avoidance task (AAT), did not support the hypothesis that brief mindful instructions would significantly mitigate the automatic approach bias toward attractive food items. This finding diverges from previous results using food images presented against a white background [14,22]. Participants demonstrated quicker approach responses than avoidance responses for both neutral and attractive foods, regardless of whether they received the mindfulness or control instructions. Additionally, response patterns to neutral food items closely mimicked the response profiles for attractive foods, with significantly slower avoidance reaction times compared to approach reaction times.

Similarly, salivary activity failed to replicate the findings of Baquedano et al. (2017) [14] and Keesman et al. (2020) [33], where brief mindfulness instruction reduced salivation in response to food images. In our study, brief mindfulness instruction did not prevent the automatic preparatory food intake reflex (CPRs) indicated by increased salivation following exposure to the advertised food stimuli. Both the mindful and control instruction groups showed comparable increases in salivation volumes from baseline (T0) to post-intervention (T2). These results suggest that automatic approach bias and salivation may be attributed to deeply ingrained and evolutionarily adaptive mechanisms that prioritize rapid approach behaviors toward high-calorie foods for survival [65]. Furthermore, the brief mindfulness instructions may be insufficient to override these automatic tendencies, especially when reinforced by the potent cues embedded in food advertising, which exploit reward-related neural pathways [66].

Baquedano et al. (2017) [14] observed that brief mindfulness instructions reduced salivation in response to food images presented against a white background. This reduction was correlated with declines in self-reported craving and increased meta-awareness, suggesting that mindfulness may enhance psychological states that, in turn, dampen autonomic responses such as salivation. In contrast, our study found that salivation and self-reported measures of hunger and food cravings increased after exposure to food cues, irrespective of the instruction manipulation. Additionally, we did not observe greater self-reported meta-awareness in the brief mindful instruction group compared to the control group. This pattern suggests that our instruction manipulation did not significantly alter participants’ physiological or psychological responses to the food stimuli.


**Neurophysiological Findings**


**Previous Findings on Approach–Avoidance Tasks:** Before addressing our findings, it is important to keep in mind that changes in the N1, N2, P3, and LPP ERP components reveal important insights into the neurophysiological processing of emotionally attractive food stimuli, particularly during approach–avoidance responses in the AAT under a standard instruction (i.e., without mindfulness intervention). The N1 component, associated with early visual and attentional processing, consistently shows greater amplitudes for emotionally salient stimuli, such as attractive foods, compared to neutral foods, reflecting their ability to capture attention due to heightened motivational and emotional salience [67,68]. The N2 component, which reflects conflict monitoring and cognitive control, demonstrates enhanced amplitudes when participants engage in avoidance responses to attractive stimuli—an indication of the effort required to override automatic approach tendencies toward emotionally salient stimuli [34,52,69]. Similarly, the P3 component, associated with the integration of motivational value, tends to elicit larger amplitudes for approach responses than for avoidance responses, particularly for attractive foods, reflecting their higher appetitive significance [39,70,71]. Finally, the LPP component, linked to cognitive reappraisal and emotional regulation, typically exhibits greater amplitudes for emotionally salient stimuli and congruent responses (e.g., approaching attractive foods) than for incongruent ones (e.g., avoiding attractive foods), reinforcing the idea that attractive stimuli evoke stronger emotional and motivational processing across multiple neural markers [39,72,73]. Together, these components suggest a dynamic interplay between early attentional capture (N1), conflict resolution (N2), motivational valuation (P3), and emotional regulation (LPP) when participants respond to emotionally attractive foods, even in the absence of specific instructional manipulation.

**Previous Findings on Brief Mindfulness Instructions in Food-Related Approach–Avoidance Tasks:** In Baquedano et al.’s research (2019) [34], brief mindful instructions were demonstrated to mitigate the automatic approach bias towards attractive food. This effect was marked by an increased N1 amplitude during the trials where participants avoided attractive food. In contrast, during the immersed condition, where the approach bias towards attractive food persisted, there were no significant differences in N1 amplitudes between approach and avoidance trials for attractive foods. For the N2 component, although there was no effect of the brief mindful instruction, the N2 amplitude was greater for attractive than for neutral food, indicating heightened cognitive control activation when participants face emotionally salient stimuli that are behaviorally relevant but must be inhibited. The P3 results indicated that mindfulness modulated attentional and motivational salience during food-related tasks. Specifically, the P3 amplitude was greater in the mindful condition than in the immersed condition, suggesting enhanced cognitive processing in the mindful condition. Additionally, approach responses elicited larger P3 amplitudes than avoidance responses, indicating stronger motivational engagement when participants moved toward food stimuli. The LPP results further emphasized the role of motivational and emotional regulation. The LPP1, LPP2, and LPP3 amplitudes were significantly greater for approach responses than for avoidance responses, reflecting heightened emotional processing during approach behavior. Furthermore, a Food type–Response interaction indicated that, for attractive foods, LPP2 amplitudes were greater for approach responses than for avoidance responses, whereas no significant differences were found for neutral foods. These findings are consistent with previous AAT literature, showing congruency effects is indexed by LPP amplitude for pleasant stimuli [51].

**Comparative Analysis of Current Findings with Prior Research**: A comparison of our findings with those of Baquedano et al. (2019) [34] revealed both alignment and divergence in the neural responses associated with approach and avoidance behaviors. In their study, mindfulness instruction appeared to modulate N1, P3, and LPP2 components. In contrast, our findings for the N1 component showed similar initial sensory processing across Instruction groups. Attractive foods elicited larger N1 amplitudes than neutral foods, reflecting the heightened attentional capture by these stimuli, but there was not an increased amplitude demonstrated in avoidance trials, a neuronal regulation that might facilitate the resolution of approach–avoidance behavioral conflicts by optimizing early perceptual processing and response selection strategies [52]. The N2 component only presented a difference in Response type, where avoidance responses elicited a greater N2 amplitude than approach responses, but there was not a Food type effect [34], indicating equal inhibition and cognitive control for both Food types. The P3 component presented only a Response type effect, wherein the amplitude for the approach response was greater than that for the avoidance response, but did not present any modulation with Instruction type. In the present study, the LPP analysis yielded interesting variations across its segments. Differently from Baquedano et al.’s research (2019) [34], the LPP2 component within this study did not reveal any significant modulation by Instruction group, Response type, Food type, or their interactions. The LPP3 component showed a significant effect of Response type, with higher amplitudes recorded for approach responses than for avoidance responses, similarly to Baquedano et al. (2019) [34]. The LPP1 ERP component showed a significant modulation by Response type, where avoidance responses presented higher amplitudes than approach responses, which was opposite to the findings of Baquedano et al. (2019) [34]. A critical variable to consider is that the decreased amplitude of LPP towards incongruent responses, compared to congruent ones, typically occurs in trials that have been successfully behaviorally regulated—specifically, when approach tendencies are not expressed behaviorally.

In Baquedano et al.’s research (2019) [34], neutral images maintained their differentiation from attractive images in terms of neural correlates, evidenced by the ‘effects’ of Food Type on N1, N2, P3, and LPPs ERPs. However, in the current study, the Food type effect was observed only in the N1 component, but not in the ERP components related to emotional response regulation, such as N2 (inhibition), P3 (integrated motivational value), and LPP (cognitive reappraisal of emotional valence). This suggests that the neurocognitive mechanisms of emotional response regulation were indistinct from those of attractive foods. This was not the case even in the immersion condition in Baquedano et al.’s research (2019) [34], where, even up to LPP, there was differentiation in the brain regarding the type of food. In summary, neurophysiological findings from ERP assessments align with prior AAT literature and are consistent with observed behavioral modulations, independent of the instruction type given. Additionally, they provide evidence supporting increased vulnerability to food advertising.

## 5. Conclusions

This study aimed to explore the effects of brief mindfulness instruction compared to control conditions on psychological, physiological, and neurophysiological responses toward advertised food cues. By incorporating advertisement-like contexts into the presentation of food stimuli, we sought to test the limits of mindfulness interventions in mitigating automatic responses to food advertisements. Comparing our results with those of Papies et al. (2012) [22] and Baquedano et al. (2017, 2019) [14,34], we found that meditation instructions lose much of their effectiveness when the stimuli are presented within a more realistic and ecologically valid context. While this does not rule out the potential of mindfulness as a tool for managing uncontrolled eating in response to external stimuli, it does suggest that brief interventions are likely insufficient to counteract the powerful influence of advertising. Testing whether more intensive mindfulness training can diminish responsiveness to attractive food stimuli should be a priority for future investigations.

These findings emphasize the deeply ingrained and socially reinforced nature of responses to food cues. Advertising appears to amplify both psychological and physiological reactivity, as seen in increased salivation and automatic approach tendencies, even toward previously neutral stimuli. While brief mindfulness instruction has demonstrated success in reducing approach biases and salivation in controlled, neutral contexts, these effects were absent in our study, where food stimuli were embedded in advertisement-like backgrounds. This discrepancy underscores the challenge posed by the powerful cues in advertising, which leverage reward-related neural pathways to drive automatic responses. Our results suggest that more prolonged or intensive mindfulness interventions may be necessary to mitigate the behavioral and physiological effects of advertising.

Another noteworthy finding is that adding advertisement-like backgrounds to food images increased the perceived attractiveness of neutral foods, effectively erasing distinctions between neutral and attractive stimuli in participants’ responses. This leads to two contrasting interpretations. On the one hand, the finding illustrates the significant power of advertising in controlling automated processes and responses, posing challenges for interventions aimed at reducing the harmful influence of advertisements for ultra-processed foods at the individual level. Consequently, regulatory actions at the collective level, such as public policy measures, are necessary. On the other hand, it opens the door to optimism: if harmful advertising can turn neutral foods into desirable ones, perhaps this same mechanism can be harnessed to promote healthier eating behaviors. Specifically, designing advertisements that make healthy foods appear more appealing could serve as a public health strategy to counteract the negative impacts of ultra-processed food advertising.

## 6. **Limitations and Future Directions**

This study has certain limitations that may affect the interpretation and generalizability of the findings. First, the mindfulness instruction was brief, consisting of a single-session intervention without prior meditation experience. While previous research has shown that such brief instructions can elicit measurable behavioral and physiological changes, it remains unclear whether a more immersive or prolonged mindfulness practice would have a greater impact. Second, the ecological validity of the intervention could be improved, not only by enhancing the realism of the images but also by incorporating more in-depth mindfulness training to strengthen participants’ engagement with the instructed mental disposition. Third, real-world food advertisements incorporate multiple persuasive factors beyond visual appeal, such as branding, emotional cues, and pricing strategies, which were intentionally excluded to maintain experimental control. While this approach allowed for a focused examination of very early automatic food-related tendencies, it limits the ecological validity of the findings, as real-world advertising involves a more complex interplay of factors. Fourth, there was an imbalance in gender and dietary preferences. While there is no prior evidence suggesting that gender or dietary preference moderates the effects of mindfulness on food-image reactivity, the possibility remains unexplored. However, given the random assignment of participants, any potential confounding effects were likely mitigated. Future research should systematically investigate whether these factors influence responsiveness to mindfulness-based interventions in food-related tasks.

Future research should address these limitations by exploring the effects of prolonged and intensive mindfulness training on food-related behaviors and physiological responses. Longitudinal designs that include multiple sessions and follow-up assessments could provide insights into the cumulative and lasting impacts of mindfulness interventions. Furthermore, it would be valuable to test this paradigm in populations with eating disorders, where mindfulness interventions may have unique therapeutic benefits. Additionally, future studies should investigate how other key advertising components, such as branding, emotional framing, and pricing strategies, interact **with** mindfulness-based interventions to strengthen or weaken food cravings. Examining these factors separately and in combination will provide a more comprehensive understanding of how food advertising influences behavior. To achieve this, future research should incorporate tasks specifically designed to assess more complex decision-making processes, allowing for the disentangling of different cognitive mechanisms involved in food-related choices. These steps will advance our understanding of how mindfulness can be effectively integrated into strategies for managing dietary behaviors and improving public health outcomes. Moreover, future studies should investigate the potential of “healthy advertising” to encourage the consumption of nutritious foods, leveraging the same psychological mechanisms that make food advertisements so effective.

## Figures and Tables

**Figure 1 brainsci-15-00240-f001:**
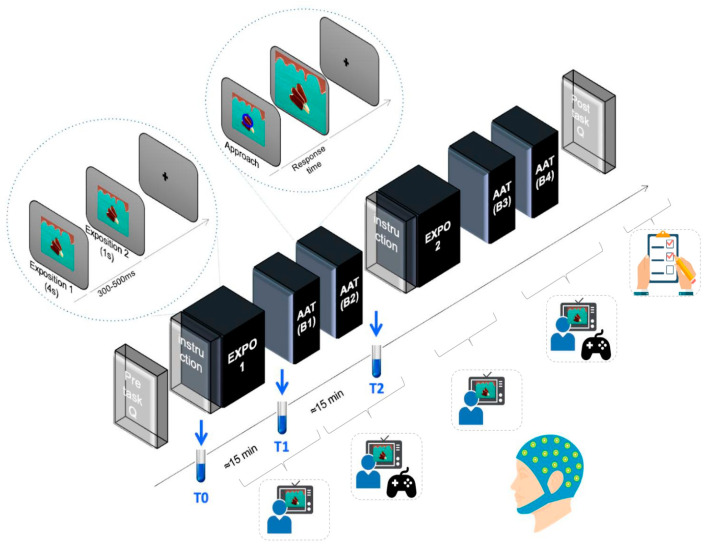
Schematic representation of the experimental procedure, which consisted of two phases, each including an instructed exposure session following specific attention instructions (Control or Mindful) and an approach–avoidance task (AAT). The experiment was divided into blocks to allow rest periods. Saliva samples were collected during the first half of the experiment.

**Figure 2 brainsci-15-00240-f002:**
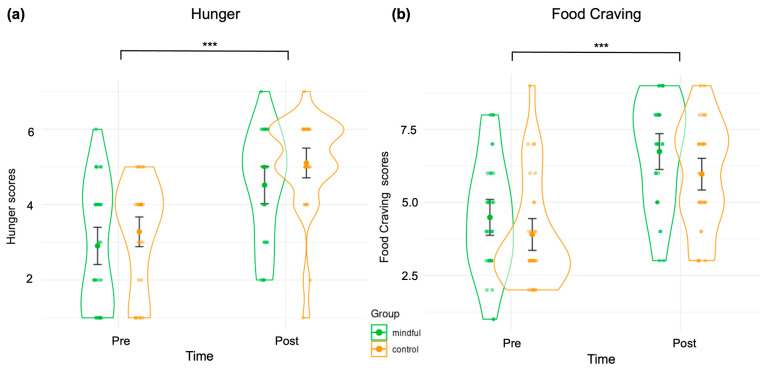
Self-report questionnaire scores: (**a**) self-perceived Hunger before (pre) and after (post) the experimental session; (**b**) self-perceived Food Craving (pre) and after (post) the experimental session. Bars denote standard error. Significances *** *p* < 0.001.

**Figure 3 brainsci-15-00240-f003:**
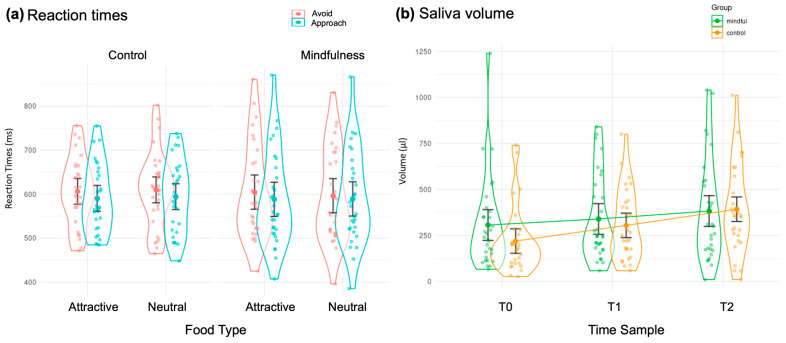
(**a**) Reaction times. RTs for approaching and avoiding attractive and neutral foods, for each of the tested instructions. The orange and cyan colors indicate the times for avoidance and approach responses, respectively. (**b**) Salivary volume. The green and pink colors indicate the Mindfulness and Control groups, respectively. T0 indicates baseline, T1 corresponds to the sample after the exposure phase, and T2 corresponds to the sample after the AAT. Bars denote standard error.

**Figure 4 brainsci-15-00240-f004:**
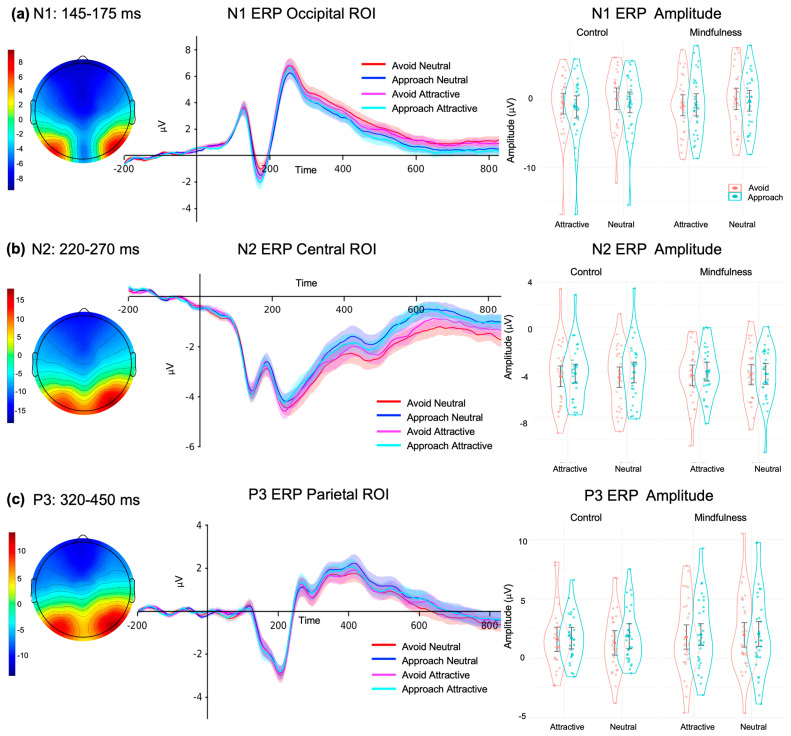
Visual ERPs, elicited by food pictures, during AAT under the two different Instruction types. (**a**) N1 ERP; (**b**) N2 ERP; (**c**) P3 ERP. **Right:** scalp plot; **middle:** ERP brainwave; **left:** quantification of the ERP amplitude in µV. Bars denote standard error.

**Figure 5 brainsci-15-00240-f005:**
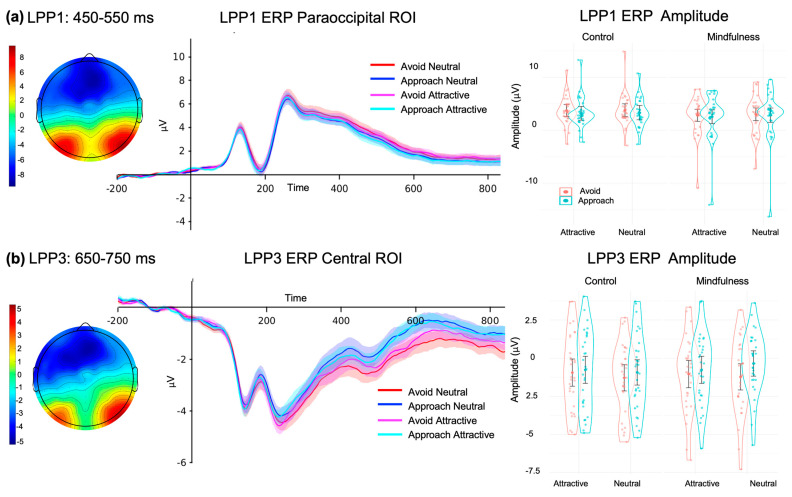
Visual ERPs elicited by food pictures during AAT under the two different Instruction types. (**a**) LPP1 ERP; (**b**) LPP3 ERP. **Right:** scalp plot; **middle:** ERP brainwave; **left:** quantification of the ERP amplitude in µV. Bars denote standard error.

**Table 1 brainsci-15-00240-t001:** Summary of trait questionnaire results and BMI measurements across Instruction groups.

Measure	Control (M ± SD)	Mindful (M ± SD)	t-Value	*p*-Value
Power of Food Scale (PFS)	57.06 ± 18.19	61.19 ± 16.46	−0.91	0.36
DEBQ—External Eating	37.58 ± 7.18	35.38 ± 7.31	1.17	0.24
Dereification-as-a-Trait	51.24 ± 4.98	48.93 ± 5.93	1.63	0.10
Body Mass Index (BMI)	25.82 ± 6.53	23.61 ± 3.1	1.65	0.10

## Data Availability

The original contributions presented in the study are included in the article; further inquiries can be directed to the corresponding author.

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
