# Peer review of "The Power of Food Advertisements: A Brief Mindfulness Instruction Does Not Prevent Psychophysiological Responses Triggered by Food Ads"

_brainsci, 2025, doi:10.3390/brainsci15030240_

Round 1
Reviewer 1 Report
Comments and Suggestions for Authors
This study investigates whether brief mindfulness instructions promoting dereification—recognizing stimuli as transient mental events—can reduce automatic consumption tendencies triggered by food advertisements. Sixty participants were divided into two groups: one receiving mindfulness instruction and the other a non-dereifying control instruction. Behavioral responses, salivation, EEG recordings, and self-reports were collected during an approach-avoidance task (AAT). Results indicated no significant differences between groups in approach behaviors, hunger, food craving, or salivation.However, following are the few suggestions and recommendations that could help to improve the overall quality of the paper:
1. Add results in quantify manner at the end of the abstract section
2. Mention the contributions of the study in Intro section
3. Structure of the paper should also be added at the end of the intro section
4. Please add a separate related work section to highlight the difference between this study and other studies
5. There should be content between heading and sub heading. please check the entire manuscript.
6. Please rename the section 2.2 with something more formal and traditional like "methodology" etc.
7. Captions of the figures are way to big. please rewrite them in maximum 1,2 lines.
8. There should be a table in the result section that summarizes the results. as tables are easy to interpret
9. Authors should compare the results of their study with existing study.
10. Outline study limitations
Comments on the Quality of English Language
There are a lot of typos and grammar mistakes. authors should proof read entire manuscript
Author Response
Comments and suggestions:
Comment 1: Add results in quantify manner at the end of the abstract section
Response 1: Thank you for this comment. We have revised the abstract to include quantitative details of the ERP results, highlighted in yellow for easy reference. Please refer to lines 36 and 37 for the updated information.
Comment 2 : Mention the contributions of the study in Intro section
Response 2: Thank you for this constructive comment. We have updated the introduction, and in the paragraph that describes the contributions of the study, we have also added a more explicit statement on how the study is expected to contribute. These additions, highlighted in yellow for easy reference, are detailed from lines 167 to 170.
Comment 3 : Structure of the paper should also be added at the end of the intro section
Response 3:Thank you for your comment. We have updated the Introduction to address this point providing a brief overview of the paper’s structure at the end of the Introduction (line 191):
"The paper follows a structured format, beginning with the Methods section, which outlines the experimental design and procedures. The Results section then presents key behavioral and physiological findings, followed by the Discussion, which analyzes our results in comparison with previous literature and their implications for food advertising and mindfulness. The paper concludes with separate sections for Conclusions, Limitations, and Future Directions, summarizing key insights, addressing constraints, and proposing avenues for future research."
However, since this follows the standard structure commonly used in cognitive neuroscience articles, we are uncertain whether this addition significantly enhances the introduction.
Comment 4: Please add a separate related work section to highlight the difference between this study and other studies
Response 4: Thank you for this suggestion. To clarify the distinction between our study and previous related work, we have restructured the Discussion section by adding subheadings to enhance clarity and explicitly address the reviewer’s concerns. The new subheadings include: Previous Findings on Approach-Avoidance Tasks (line 664), Previous Findings on Brief Mindfulness Instructions in Food-Related Approach-Avoidance Tasks (line 688), Comparative Analysis of Current Findings with Prior Research (line 712). We appreciate the reviewer’s feedback, as these additions improve the organization of the discussion and better contextualize our findings within the existing literature.
Comment 5: There should be content between heading and subheading. please check the entire manuscript.
Response 5: We thank the reviewer for this comment. We have identified three instances where a heading and subheading appeared without content in between. To address this, we have adjusted the formatting to ensure clarity and prevent missing content. These modifications are highlighted in yellow at lines 300 and 316, as well as at lines 332, 335, 366, 474, and 484.
Comment 6: Please rename the section 2.2 with something more formal and traditional like "methodology" etc.
Response 6: We thank the reviewer for pointing this out. We have now renamed Section 2.2 as "Experimental Procedure" to ensure a more formal and traditional title. This change is highlighted in yellow at line 216.
Comment 7:. Captions of the figures are way to big. please rewrite them in maximum 1,2 lines.
Response 7:We appreciate the reviewer’s suggestion. We identified that the captions for Figures 1 and 3 were too long and have significantly condensed them to a maximum of one to two lines. The revised captions are highlighted in yellow at line 245 for Figure 1 and line 547 for Figure 3.
Comment 8: There should be a table in the result section that summarizes the results. as tables are easy to interpret
Response 8:We acknowledge the reviewer’s suggestion and agree that presenting the trait self-report results in a table improves clarity and readability. Therefore, we have extracted the values from the paragraph and formatted them into a table for easier interpretation. The full section has been highlighted in yellow, starting at line 475.
Comment 9: Authors should compare the results of their study with existing study.
Response 9:Thank you for your comment. This comparison was already included in the Discussion section; however, it may not have been explicitly clear. To improve clarity, we have now made this comparison more explicit and structured it under the subheading "Comparative Analysis of Current Findings with Prior Research"(line 712). We appreciate the reviewer’s feedback, as this adjustment enhances the accessibility of our discussion.
Comment 10:. Outline study limitations
Response 10:To address this point, we have clearly separated the paragraph discussing limitations from the one outlining future directions. Additionally, we have enumerated the limitations to improve clarity and ensure they are explicitly stated. The revised Limitations section can be found at line 790, highlighted for easy reference
Comment 11: Comments on the Quality of English Language. There are a lot of typos and grammar mistakes. authors should proof read entire manuscript
Response 11: We appreciate the reviewer’s feedback. We have carefully proofread the entire manuscript, correcting typos and grammar mistakes to improve the overall quality of the English language.
Reviewer 2 Report
Comments and Suggestions for Authors
This study investigated whether brief mindfulness instructions are enough to prevent psychophysiological responses induced by food ads. The findings showed brief mindfulness instructions were not strong enough to shield participants from automatic responses elicited by food ads. I think the theme is interesting and the paper is well-written in general, but the manuscript still needs improvement.
The introduction connected food ads and mindfulness intervention well. However, I felt that the introduction does not fully emphasize how advertising might strengthen cravings beyond just food images. In addition, what factors in food ads can make a difference? I think treating food ads as a uniform category may be misleading. It should explain specific mechanisms through which cravings are amplified.
I think having BMI > 30 as an exclusion criterion is reasonable. However, was extremely low BMI also considered?
Essential technical details are missing.
List all EEG channels.
Reveal the specification of the filter used, like filter type, window type.
The authors stated that defective channels were identified by eye inspection and omitted from the preprocessing steps. Please describe the specific criteria used for manual identification.
What algorithm was used for ICA? Reveal all the relevant parameters as well.
Line 358: What does the ‘score’ exactly mean here? There are several criteria to reject components. Reveal exact quantities used and criteria.
As far as I know, a head model is required to re-reference to infinity. Reveal all relevant information.
I am confused about how you handled data rank, and the current description seems unclear. According to the official EEGLAB documentation, spherical channel interpolation reduces the effective data rank, which can impact ICA. Since you used EEGLAB, you likely used ‘runica’ for ICA. If you applied the PCA option, you could clarify this by stating something like 'We used the PCA option to retain the full data rank [a relevant paper].' Alternatively, if you performed channel interpolation after ICA, which avoids rank deficiency issues for ICA, you should explicitly mention it with a statement like 'We ran ICA before channel interpolation, ensuring that the data retained full rank [a relevant paper].' Please refer to the official EEGLAB reference for details.
https://sccn.ucsd.edu/wiki/Makoto's_preprocessing_pipeline#A_study_on_the_ghost_IC_was_published_.28added_on_04.2F04.2F2023.29
Similar to what I pointed out in the introduction, the findings that mindfulness didn’t reduce cravings in this study suggest that food advertising might influence cravings in ways beyond just showing appealing food images. Unlike plain food pictures, ads often include emotional cues, familiar branding, and persuasive elements that could reinforce automatic approach tendencies. A more detailed analysis of these factors is necessary to understand why mindfulness may not counteract food ads as effectively as it does with plain food images.
Author Response
Comment 1: The introduction connected food ads and mindfulness intervention well. However, I felt that the introduction does not fully emphasize how advertising might strengthen cravings beyond just food images. In addition, what factors in food ads can make a difference? I think treating food ads as a uniform category may be misleading. It should explain specific mechanisms through which cravings are amplified.
Response 1: We fully acknowledge the reviewer’s insightful comments regarding the multifaceted nature of food advertising and its influence on cravings. We recognize that advertisements employ various techniques beyond visual appeal, such as brand associations, emotional cues, multisensory elements, and pricing strategies, all of which engage distinct neurocognitive mechanisms to promote food consumption. For example, pricing and branding can affect perceived value and desirability, leading to complex cost-opportunity decision-making processes.
In the case of our study, we specifically focused on modifying the visual imagery of food advertisements to enhance subjective realism, while intentionally excluding other advertising features that recruit higher-order cognitive processes, such as branding, discounts, and contextual associations. This decision was made to isolate the effects of visual exposure on automatic approach tendencies, ensuring that the experimental design remained aligned with the measurement of early, automatic food-related responses, rather than more complex, fractionated decision-making mechanisms. A paradigm designed to investigate later-stage cognitive processes would require a different methodological approach.
To address this concern raised in Comment 1 and 11, but without losing the focus of the study, we have now explicitly mentioned other advertising factors that influence food cravings in the Introduction ( lines 50 -53)while maintaining our scope on the visual aspects of advertising that drive subjective realism. Later in response to Comment 11, we acknowledge in the Limitations and Future Directions section that real-life advertisements involve multiple interacting factors. We highlight that future studies should systematically examine each of these elements separately, using cognitive paradigms designed to investigate more complex decision-making processes.(lines 799-804 and lines 817 -824)
Comment 2:. I think having BMI > 30 as an exclusion criterion is reasonable. However, was extremely low BMI also considered?
Response 2: Thank you for your comment. In addition to excluding participants with a BMI > 30, we also set a minimum BMI cutoff of 18.5 to ensure that only individuals with a normal BMI range were included. We have now added this parameter to the Methods section, specifically in line 206, and highlighted it for clarity.
Comment 3: Essential technical details are missing.
Response 3: Thank you for pointing this out. We address the specific technical details in the following comments and will incorporate them into the text where pertinent.
Comment 4: List all EEG channels.
Response 4: Thank you for your request. The EEG recording was conducted using a BioSemi 128-channel system with an ABC layout, including electrodes A1–A32, B1–B32, C1–C32, and D1–D32. More details on the electrode placement can be found at (https://www.biosemi.com/headcap.htm) Given the standardization of this layout, we consider it unnecessary to list all individual electrodes in the article, but we are happy to provide additional details if required.
Comment 5: Reveal the specification of the filter used, like filter type, window type.
Response 5:Thank you for your request. The EEG data was high-pass filtered at 0.1 Hz using an IIR Butterworth filter with a 12 dB/oct roll-off. The low-pass filter was applied at 30 Hz, also using an IIR Butterworth filter with a 12 dB/oct roll-off, both were implemented in ERPLAB. We have added these filter specifications to the Methods section, specifically in line 345.
Comment 6: The authors stated that defective channels were identified by eye inspection and omitted from the preprocessing steps. Please describe the specific criteria used for manual identification.
Response 6:Thank you for your request. Defective channels were identified based on the following criteria: flat channels, channels exhibiting sudden voltage jumps (~10 ms) exceeding ~50 μV in amplitude, and channels with signal fluctuations beyond ~100 μV. These criteria ensured the exclusion of artifacts while preserving data quality.
Comment 7: What algorithm was used for ICA? Reveal all the relevant parameters as well.
Response 7:Thank you for your request. We used the Extended Infomax algorithm (runica) for ICA decomposition, with no PCA dimension reduction, as implemented in EEGLAB. All parameters were set to EEGLAB's default settings to ensure standardization and reproducibility. We have now added the specification of the algorithm in the Methods section, specifically in line 350, and highlighted it in yellow for clarity.
Comment 8: Line 358: What does the ‘score’ exactly mean here? There are several criteria to reject components. Reveal exact quantities used and criteria.
Response 8: Thank you for your request. The term "score" refers to the independent component (IC) class probability assigned by the ICLabel classifier (Pion-Tonachini et al., 2019). This probability score was used to categorize components based on their likelihood of representing neural or artifact-related activity. We have now clarified this in the manuscript at line 352 and highlighted the revision in yellow for transparency.
Comment 9: As far as I know, a head model is required to re-reference to infinity. Reveal all relevant information.
Response 9:Thank you for your request. The three-concentric-sphere head model was used for re-referencing to infinity, following the methodology described in Dong et al. (2017). Given the extensive details provided in their work, we consider it too specific to include in the manuscript but will reference Dong et al. (2017) in line 355, to ensure readers can access the full methodological explanation.
Comment 10:. I am confused about how you handled data rank, and the current description seems unclear. According to the official EEGLAB documentation, spherical channel interpolation reduces the effective data rank, which can impact ICA. Since you used EEGLAB, you likely used ‘runica’ for ICA. If you applied the PCA option, you could clarify this by stating something like 'We used the PCA option to retain the full data rank [a relevant paper].' Alternatively, if you performed channel interpolation after ICA, which avoids rank deficiency issues for ICA, you should explicitly mention it with a statement like 'We ran ICA before channel interpolation, ensuring that the data retained full rank [a relevant paper].' Please refer to the official EEGLAB reference for details.https://sccn.ucsd.edu/wiki/Makoto's_preprocessing_pipeline#A_study_on_the_ghost_IC_was_published_.28added_on_04.2F04.2F2023.29
Response 10: Thank you for your comment. As stated in the manuscript, ICA was performed before channel interpolation to ensure that the data retained full rank. This is explicitly outlined in the preprocessing section, where we describe the order of steps taken. We agree that interpolating channels before ICA would lead to rank deficiency, and we carefully followed best practices to avoid this issue.
To further clarify this point, we have now explicitly stated that ICA was performed prior to interpolation in the Methods section (line 349). Additionally, we acknowledge the importance of referencing relevant literature on this matter and have now cited Kim et al. (2023) to reinforce the reasoning behind this approach. We appreciate the reviewer’s attention to methodological rigor and have taken steps to ensure this is clearly conveyed in the revised manuscript.
Kim, H., Luo, J., Chu, S., Cannard, C., Hoffmann, S., & Miyakoshi, M. (2023). ICA’s bug: How ghost ICs emerge from effective rank deficiency caused by EEG electrode interpolation and incorrect re-referencing. Frontiers in Signal Processing, 3, 1064138. https://doi.org/10.3389/FRSIP.2023.1064138/BIBTEX
Comment 11: Similar to what I pointed out in the introduction, the findings that mindfulness didn’t reduce cravings in this study suggest that food advertising might influence cravings in ways beyond just showing appealing food images. Unlike plain food pictures, ads often include emotional cues, familiar branding, and persuasive elements that could reinforce automatic approach tendencies. A more detailed analysis of these factors is necessary to understand why mindfulness may not counteract food ads as effectively as it does with plain food images.
Response 11: Again we appreciate the reviewer’s insightful comment. This concern has been addressed in Response to Comment 1, where we discuss the multifaceted nature of food advertising and its influence on cravings beyond visual appeal. Specifically, we acknowledge that advertisements incorporate emotional cues, branding, and persuasive elements, which could reinforce automatic approach tendencies. We have now explicitly included these factors in the Introduction (line 50) and further expanded on them in the Limitations and Future Directions section, where we highlight the need for future research to systematically examine these additional factors and their interaction with mindfulness-based interventions. (lines 799 to 804 to 800 and 817 to 824)
Reviewer 3 Report
Comments and Suggestions for Authors
The authors present an elaborate and interesting article titled: “The power of food advertisements: A brief mindfulness instruction does not prevent psychophysiological responses triggered by food ads.” The manuscript is generally well-written and provides a thorough description of the research approach. However, I have a few comments and suggestions that should be considered.
Comments and suggestions:
- In the abstract, the abbreviations LPP and ERP are not formally introduced. Moreover, the EEG components N1, N2, and P3 are mentioned, but their meaning/theoretical basis is not immediately clear without the information provided in the Methods section (Section 2.7.2). A brief addition would make it easier for the reader to understand.
- The introduction and methods section are described in great detail. Methodologically, I particularly appreciate the conducted sample size calculation.
- Some of the figures are quite small. Increasing their resolution and size would enhance visibility and readability.
- The study population raises interesting discussion points. There is an imbalance in the gender distribution (42 females and 18 males), as well as between vegetarians, vegans and meat-eaters (10/3/47 participants). It would be valuable to discuss how these factors, in particular, might influence the study results.
- I appreciate the authors’ clear statement on the study’s limitations and suggestions for future research.
- In section Appendix A, I recommend adding a brief note indicating that the linked videos are in Spanish, not in English.
Author Response
Comment 1: In the abstract, the abbreviations LPP and ERP are not formally introduced. Moreover, the EEG components N1, N2, and P3 are mentioned, but their meaning/theoretical basis is not immediately clear without the information provided in the Methods section (Section 2.7.2). A brief addition would make it easier for the reader to understand.
Response 1: Thank you for pointing this out. We have now added the full terms for "ERP" and "LPP" in the abstract (see lines 31 and 35) to ensure clarity. Regarding the theoretical basis of the N1, N2, and P3 ERP components, we had previously introduced N1 and LPP in the Introduction, and we have now added a brief explanation of P3, given its relevance to mindfulness-related modulations (lines 130 to 134). Since N2 was not previously introduced in the Introduction—as it has not been shown to be modulated by mindfulness during the approach-avoidance task—we have included a brief clarification in the Results section to ensure its meaning is clear when interpreting the findings. (lines 574 to 576).
Comment 2:The introduction and methods section are described in great detail. Methodologically, I particularly appreciate the conducted sample size calculation.
Response 2: We sincerely thank the reviewer for their positive feedback. We appreciate the acknowledgment of the detailed description of the Introduction and Methods sections, as well as the conducted sample size calculation.
Comment 3: Some of the figures are quite small. Increasing their resolution and size would enhance visibility and readability.
Response 3: Thank you for pointing this out. We have increased the resolution and font size of the figures to enhance visibility and readability.
Comment 4: The study population raises interesting discussion points. There is an imbalance in the gender distribution (42 females and 18 males), as well as between vegetarians, vegans and meat-eaters (10/3/47 participants). It would be valuable to discuss how these factors, in particular, might influence the study results.
Response 4:We acknowledge that our sample was imbalanced in terms of gender (with a male-to-female ratio of 1:2.3) and dietary preferences (with a non-meat-to-meat eater ratio of 1:3.6). This imbalance could present challenges in interpreting the findings if responses to food images under mindfulness and control conditions varied by gender or dietary preference, particularly if these imbalances were unevenly or systematically distributed across experimental and control groups. However, we found no prior research suggesting that gender or dietary preference moderates the effects of mindfulness versus control instructions on food-image reactivity. Moreover, even if these factors did influence image reactivity or instructional effects, their random distribution across groups likely mitigated any potential confounding effects. Nonetheless, given the absence of systematic studies exploring these possible differences and moderation effects, future research should address this gap. We have added these comments to the Discussion under the Limitations and Future Directions subsection. (lines 804 to 809)
Comment 5: I appreciate the authors’ clear statement on the study’s limitations and suggestions for future research.
Response 5:We sincerely thank the reviewer for their positive feedback. We appreciate the acknowledgment of our efforts in clearly stating the study’s limitations and proposing directions for future research.
Comment 6: In section Appendix A, I recommend adding a brief note indicating that the linked videos are in Spanish, not in English.
Response 5:Thank you for pointing this out. We have added a note in Appendix A indicating that the video’s audio is in Spanish.
Round 2
Reviewer 1 Report
Comments and Suggestions for Authors
Authors have addressed all the comments
Comments on the Quality of English Language
English is sufficient
Author Response
Comment: Authors have addressed all the comments and the manuscript can be accepted in current form
Response: Thank you for your positive feedback and for confirming that all comments have been addressed satisfactorily. We appreciate the time and effort the reviewers have dedicated to improving our manuscript.
Reviewer 2 Report
Comments and Suggestions for Authors
I appreciate the authors’ response. While the response is satisfactory, some minor clarification is still needed.
Response 8: Thank you for your request. The term "score" refers to the independent component (IC) class probability assigned by the ICLabel classifier (Pion-Tonachini et al., 2019). This probability score was used to categorize components based on their likelihood of representing neural or artifact-related activity. We have now clarified this in the manuscript at line 352 and highlighted the revision in yellow for transparency.
- In the main text, the authors stated, “between 0.8 and 1 in the aforementioned artifactual categories were removed”. Does this mean that components with a probability below 0.8 were not rejected, even if identified as artifacts (e.g., 79% muscle component)? I have often seen cases where only brain components with a probability above a certain threshold (e.g., 80%) were retained, so I wanted to confirm whether the description in the text is correct. I am not criticizing, just seeking clarification. If needed, please update the main text accordingly.
Response 9:Thank you for your request. The three-concentric-sphere head model was used for re-referencing to infinity, following the methodology described in Dong et al. (2017). Given the extensive details provided in their work, we consider it too specific to include in the manuscript but will reference Dong et al. (2017) in line 355, to ensure readers can access the full methodological explanation.
- It is completely fine to cite a reference without explaining everything. However, for reproducibility, the authors should at least mention key details, such as the model used. Please update the main text accordingly. You might want to add it like “… , using the three-concentric-sphere head model.”
Response 10: Thank you for your comment. As stated in the manuscript, ICA was performed before channel interpolation to ensure that the data retained full rank. This is explicitly outlined in the preprocessing section, where we describe the order of steps taken. We agree that interpolating channels before ICA would lead to rank deficiency, and we carefully followed best practices to avoid this issue.
To further clarify this point, we have now explicitly stated that ICA was performed prior to interpolation in the Methods section (line 349). Additionally, we acknowledge the importance of referencing relevant literature on this matter and have now cited Kim et al. (2023) to reinforce the reasoning behind this approach. We appreciate the reviewer’s attention to methodological rigor and have taken steps to ensure this is clearly conveyed in the revised manuscript.
- Again, you should provide sufficient and essential information in the main text. i.e., here you should clarify how the rank issue was handled.
- I suggest revising as follows:
“At this point defective channels were spherically interpolated [50].” -> “At this point, defective channels were spherically interpolated subsequent to ICA to retain a full rank for ICA [50].”
Author Response
Comment 8.R2: In the main text, the authors stated, “between 0.8 and 1 in the aforementioned artifactual categories were removed”. Does this mean that components with a probability below 0.8 were not rejected, even if identified as artifacts (e.g., 79% muscle component)? I have often seen cases where only brain components with a probability above a certain threshold (e.g., 80%) were retained, so I wanted to confirm whether the description in the text is correct. I am not criticizing, just seeking clarification. If needed, please update the main text accordingly.
Response 8.R2: Thank you for your inquiry regarding the artifact rejection criteria used in our study. To confirm, components identified as artifacts with a probability score between 0.8 and 1 were indeed removed. This means components with a probability below 0.8, even if identified as artifacts, were not automatically rejected. We believe the text reflects our methodology, but we appreciate your vigilance in ensuring the clarity and precision of our descriptions.
Comment 9.R2: it is completely fine to cite a reference without explaining everything. However, for reproducibility, the authors should at least mention key details, such as the model used. Please update the main text accordingly. You might want to add it like “… , using the three-concentric-sphere head model
Response 9.R2: Thank you for your suggestion regarding the inclusion of key details for reproducibility. We have updated the manuscript as recommended. Please see line 356, where we now specify the model used: "Data was re-referenced to infinity, using the three-concentric-sphere head model, with the REST 1.2 toolbox [51]."
Comment 10.R2: Again, you should provide sufficient and essential information in the main text. i.e., here you should clarify how the rank issue was handled.
- I suggest revising as follows:
“At this point defective channels were spherically interpolated [50].” -> “At this point, defective channels were spherically interpolated subsequent to ICA to retain a full rank for ICA [50].”
Response 10.R2: Thank you for your suggestion regarding the description of our data preprocessing steps. We have revised the manuscript to include your recommendation for clarity on the handling of defective channels. Specifically, on line 355, the text has been updated to state: "Defective channels were spherically interpolated after performing ICA to maintain the data's full rank for accurate ICA computation [50]."